# The Influence of Focus on the Activation of Alternatives in Speech Production—An Online Picture-Word-Interference Experiment

Beate Bergmann [1,*], Yanru Lu [1,*]  and Katharina Spalek [1,2,*] 

1   Institut für deutsche Sprache und Linguistik, Humboldt-Universität zu Berlin, 10099 Berlin, Germany
2   Institut für Linguistik, Heinrich-Heine-Universität, 40225 Düsseldorf, Germany
*   Correspondence: beate@beatebergmann.com (B.B.); yanru.lu@hu-berlin.de (Y.L.);
    katharina.spalek@hhu.de (K.S.)

**Abstract:** In previous picture naming tasks, semantically related distractor words (co-hyponyms to the target word) induce interference, which is usually taken as evidence for lexical competition. In an online picture-word-interference experiment, we showed that distractor words that share a feature with the target (here: their natural prototypical color), also induced interference. Pictures were not named with single words but with short descriptive sentences ("The heart is red"). Focus on the noun modulated the interference effect. In particular, when target and distractor were presented simultaneously, the interference effect was significantly reduced in the narrow focus condition, compared to broad focus. We discuss our findings for focus production against the findings on language comprehension reported in the literature, which mostly observed facilitatory effects of focus marking on the comprehension of focus alternatives.

**Keywords:** focus production; focus alternatives; picture-word interference; semantic interference; online experiments

## 1. Introduction

A speaker can describe the same situation in different ways—she can choose different words, order these words differently, or produce the same words in the same order, but with different prosody. Some of these choices depend on the information structure of a sentence. For example, if the speaker produces example (1), with a contrastive accent on NEXT DOOR,[1] she indicates that alternatives to the accented (the focused) constituent are relevant for the interpretation of the utterance (see Krifka 2008; Rooth 1992). Therefore, a listener might infer that there were other, less helpful, neighbors. In the following, we will refer to constituents that can replace the focused constituent in the current utterance—such that the resulting sentence is still grammatical and meaningful—as FOCUS ALTERNATIVES.

(1)    The neighbor NEXT DOOR helped with the groceries.

There is substantial evidence that focus alternatives become activated during language comprehension (see Section 1.4). Here, we investigate whether a SPEAKER who produces an utterance with a focused element, chooses the word for this referent from a set of alternatives. A first study from our laboratory (Bergmann and Spalek 2022, Exp. 3), using primed cross-modal lexical decision during language production, suggests this is the case (see Section 1.5 for more details). In the present study, we use a different experimental paradigm, the picture-word-interference (from here onwards: PWI) paradigm, to further investigate this question.

The current article begins with a brief overview of the theoretical and experimental background; first, on language production, and second, on information-structural focus, before we outline our experiment in detail. We will then discuss the results in light of focus alternatives as well as speech production research.

*1.1. Lexical Access during Language Production*

Models of speech production (Abdel Rahman and Melinger 2009, 2019; Caramazza 1997; Dell 1986; Levelt et al. 1999) assume that it is a staged process. In a first step, a lexical representation (called "lemma" in Levelt 1989; Levelt et al. 1999) matching the speaker's idea (the "message", according to Levelt 1989), is selected from a semantically ordered store of representations. The lemma is the entry point for a phonological representation (the word form), which can then be passed on to the articulators. This is a very superficial description, glossing over a lot of detail. One aspect in which the models differ concerns whether or not lexical access is a competitive process. A number of models assume that both the target lexical representation and also the semantically related words are active, and the target has to be selected from this set of co-active representations. The stronger the activation of the competitors, the more difficult it is to select the target (e.g., Caramazza 1997; La Heij et al. 2006; Levelt et al. 1999; Roelofs 1992; for a different view, see e.g., Finkbeiner and Caramazza 2006; Janssen et al. 2008; Mahon et al. 2007).

*1.2. Empirical Evidence for Lexical Competition*

Evidence for competitive lexical selection processes comes from various experimental paradigms, such as picture-word interference (PWI) (e.g., Damian and Bowers 2003; Schriefers et al. 1990), or blocked naming (e.g., Belke et al. 2005). In a classic PWI task, a speaker is asked to name an object presented as a simple line drawing ("target"), and naming latencies are measured. A so-called distractor, presented visually (e.g., as a superimposed word or picture) or auditorily, appears together with the picture. Participants are asked to ignore the distractor during the naming task. Despite the instructions, participants are never fully able to ignore the distractor. The relationship between distractor and target, as well as the relative timing of their presentation, affect the naming latencies. Since the aim of a PWI task is to investigate (unconscious) planning processes prior to speaking, words are usually not presented later than 450 ms after picture onset (naming simple pictures usually takes at least 600 ms before participants start speaking). Glaser and Düngelhoff (1984) systematically varied stimulus onset asynchronies (SOAs) from −400 ms (i.e., the distractor was presented 400 ms before the picture appeared) to 400 ms (i.e., the distractor was presented 400 ms after the picture), in steps of 100 ms. The authors found interference for targets and distractors with a common hypernym (e.g., FRUIT), that was strongest between −100 ms and 100 ms.

Most models of language production assume that we start accessing the mental lexical, given a conceptual representation which is provided by the picture in classic PWI tasks. The lexicon is built as a network of nodes that are linked by virtue of their semantic relation. For example, the co-hyponyms *apple* and *pear* are both linked to the hyperonym FRUIT. Due to spreading activation along those links, related concepts are also active when we produce a word. If the wrong concept accumulates too much activation, this results in a speech error such as "Would you like an apple, uh, I mean, a pear?".

Roelofs (1992) assumes that the distractor word primes the distractor lemma (but also the target lemma), just as the target picture primes not only its lemma, but also semantically related lemmas. In this constellation, it takes longer for the target lemma to become the most highly activated one, as opposed to target lemma in a constellation where the distractor word does not prime a member of the target's lexical cohort. Abdel Rahman and Melinger (2009) point out that, for interference to occur, it is crucial that a cohort of lexical items is involved. That is, target and distractor do not just prime each other, but each of them sends activation to more related words, for example *plum*, *strawberry*, and *cherry* in the example just discussed. If the relationship between target and distractor is a one-to-one relationship as in, for example, associative relations (*bee—honey*) or part-whole relations (*car–wheel*), then the priming from distractor to target will not be offset by interference due to a co-activated lexical cohort, and therefore, facilitation will be observed (e.g., Alario et al. 2000; Costa et al. 2005). Abdel Rahman and Melinger (2009) show that lexical competition can be induced

even for associatively related items, so long as they are presented repeatedly in a blocked naming paradigm.

To sum up, interference effects in PWI are a hallmark for lexical competition.

### 1.3. Focus Marking and Interpretation

Focus can be marked syntactically, morphologically, or prosodically. In German, as in English, prosodic focus marking is the most common. Focus in German is typically marked by increases in prominence-lending parameters such as duration, pitch peak, and pitch excursion, and by hyperarticulation on the accented syllables in focused words (Baumann et al. 2007; Baumann et al. 2006). The introduction of new items that are not contrastive is often reflected in the use of a high (H*) pitch accent, whereas contrastive elements are produced with a lower starting point (L+H*) (Baumann et al. 2000; Féry and Kügler 2008). However, not all speakers in language production studies show the expected categorical shift from H+!H* to H* to L+H* for broad, narrow, and contrastive focus, respectively (Grice et al. 2017).

For interpretation, too, the domains of pitch accents partly overlap: Watson et al. (2008) report an eye-tracking study (for English) where participants listened to spoken instructions and had to manipulate objects presented on a display. While stimuli spoken with an L+H* accent strongly biased listeners' anticipations to a contrastive referent, H* accents were compatible both with contrastive and new referents.

### 1.4. Focus Effects during Comprehension

Cross-modal priming studies have shown that focus alternatives are activated during language comprehension. Braun and Tagliapietra (2010) presented auditory sentences in Dutch with contrastive focus on a critical noun, for example: "My son likes SPINACH.[2]" These sentences were followed by a letter string on the screen and participants had to decide whether the letter string was a word or not (lexical decision). The authors observed that participants were faster to respond to a related word (e.g., *kale*) than to an unrelated word, but only when the critical word in the sentence had been produced with a contrastive accent. Braun and Tagliapietra conclude that the accent evokes alternatives, and that these alternatives are therefore recognized faster in the lexical decision task (see also Yan and Calhoun 2019, for similar findings in Mandarin Chinese).

Husband and Ferreira (2016) also used cross-modal priming and lexical decision in English, to investigate the time course of alternative activation. They presented an alternative to the contrastively related word, a related word that cannot be an alternative, or an unrelated word. The word was presented in written form, either directly after the contrastively focused word had been spoken, or with a delay of 750 ms. The prime word in the sentence was either spoken with contrastive focus or broad focus. The key finding concerns the behavior of alternatives and non-contrastively related words in the contrastive focus condition. With immediate presentation, both types of related words were primed relative to the control condition. However, with a delay of 750 ms, only the alternative was primed (in the contrastive focus condition). The authors argue that, initially, activation spreads to all related words. As time passes, contrastive focus triggers a selection mechanism such that only focus alternatives remain active, whereas non-alternatives become de-activated.

Braun et al. (2019) and Braun and Biezma (2019) used eye tracking to show alternative activation in German. While participants heard a spoken sentence, four words were presented on a computer screen. The authors investigated whether and in which conditions participants looked at an alternative to the contrastively focused sentence object. With this paradigm, they could show yet again that a contrastive accent activates alternatives, but that a narrow focus accent does not.

To sum up, there is evidence for the activation of focus alternatives during comprehension. The findings for production are much scarcer, and will be discussed in the next section.

*1.5. Focus Effects during Production*

Little is known about focus alternatives in speech production. Studies on focus production come mainly from focus particle research in language acquisition (e.g., Höhle et al. 2009; Müller et al. 2009). Höhle et al. (2009), for instance, investigated spontaneous and elicited speech with the focus particle *auch*, which means 'also' in German, in children aged 1 to 4 years. Results suggest a delayed production for unaccented *auch* compared to accented *auch*, as well as a discrepancy in linguistic skills of performance and competence. A different strand of research has investigated how the presence of contrast in the surroundings affects articulation (e.g., Ito and Speer 2008, for English).

However, to our knowledge, alternatives to a focused element are neglected in production studies in language acquisition research.

A very recent contribution to focus production in German comes from Bergmann and Spalek (2022). In a series of experiments, they demonstrated that speakers do activate focus alternatives. In Experiment 3, a picture naming task, participants were asked to name pictures of objects and animals in a specific color (e.g., a white sheep). Items were introduced prior to the experiment such that participants could familiarize themselves with the item sets (i.e., alternative sets). These sets were categorized according to the feature of 'most prototypical natural color' (e.g., *elephant*, *key*, *stone*, *can*, *donkey*, and *nail* for the set of gray-colored objects and animals). The way the experiment was set up, the items were possible alternatives, because the to-be-produced sentences had the form, "The X is color y", and therefore every element of a color set could be inserted for X in this context. In the following, we will sometimes reference contextual relatedness, but more often color relatedness, since this is how relatedness was operationalized not only in Bergmann and Spalek (2022: Exp. 3), but also the present study.

Target pictures were preceded by context pictures, in order to induce contrastive intonation in the naming process. In the "narrow focus condition", a speaker responds to the context picture with, "The tooth is white", and to the target picture with, "The sheep is white". In this case, both the context and the target pictures are elements from the set of white things. The target picture there is an alternative to the previously named picture. By contrast, in the broad focus condition, a speaker would say, for example, "The nest is brown", followed by, "The sheep is white", that is, no contrast was induced.[3] For target pictures, participants had to perform a lexical decision task (henceforth: LDT) on a written letter string ("probe word"), before naming the picture. When a word is presented while the participant is still preparing to name the picture, properties of the picture name will affect the LDT (see Levelt et al. 1991, for a demonstration of this). In the study by Bergmann and Spalek (2022), the letter string was the name of another object sharing the color feature with the target, (e.g., *igloo*). Results revealed that participants were slower in the LDT when the picture was to-be-named with contrastive focus, as opposed to when it was to-be-named with broad focus. In the former, but not the latter case, the probe word is a potential alternative to the target word. We concluded that the longer LD-times reflected increased lexical competition, because the focused element had activated all members from its alternative set (here the set of white items), which interfered with the lexical decision on one of these alternatives.

In summary, there is some evidence that speakers activate focus alternatives in their mind. We want to replicate this, and at the same time learn more about the time course of alternative activation during language production.

To do so, we chose the PWI paradigm. It is well known that co-hyponyms cause semantic interference in PWI tasks (e.g., Damian and Bowers 2003; Schriefers et al. 1990) when they are presented simultaneously or near-simultaneously with the target. To reiterate: The most common explanation of these effects is that, when we try to access a lexical element given the concept provided by the picture, related lemmas became activated due to activation spreading in the mental lexicon. If one of these competitors is further activated by virtue of the distractor, choosing the correct lemma becomes more difficult. This is reflected in increased naming latencies. In the same vein, we argue that, if alternatives

are activated when speakers retrieve a focused target lemma, then presenting one of these alternatives as distractor in a PWI task, will increase naming latencies. In the current experiment, just as in Experiment 3 of Bergmann and Spalek (2022), we use sets of items (e.g., *salad*, *caterpillar*, *cactus*) that are related based on a feature that they share, which in this case was their common prototypical natural color.

With the current online PWI experiment, we therefore explore: (1) whether we can replicate the finding that focus alternatives are activated in language production; (2) whether we can replicate the finding that alternative sets can be based on contextual relatedness in the form of a shared property; and (3) whether we can gain a better understanding of the time course of focus alternative activations, by varying the relative timing of target and distractor presentation.

## 2. The Current Study

### 2.1. Aims and Hypotheses

With the current experiment, we explore focus alternative activation during speech *production*. We were interested in lexical selection during the speech planning phase, i.e., before speech onset. Unlike the usual picture naming task that requires a single word response (e.g., 'apple' for a picture that shows an apple), the task was modified such that participants had to answer in a complete sentence in order to induce contrastive intonation (see Section 2.2.2). If focus alternatives are co-activated during lexical selection in focus production, and assuming that co-activated elements/concepts are in competition, then this should be reflected in the results. In other words, we would expect to find a stronger interference effect in the PWI task (see e.g., Damian and Bowers 2003) with the narrow focus condition (see below for details), rather than with the broad focus condition. To clarify: reaction times should be longer in the narrow focus condition if the object referred to by the superimposed distractor word has the same color as the picture name, than if it has a different color. The reaction time difference for targets and distractors, with or without color overlap, should be absent (or at least smaller) in the broad focus condition.

We used the PWI paradigm with a range of stimulus onset asynchronies (SOAs): −100 ms, 0 ms, and 100 ms. These are the SOAs for which semantic interference in PWI has most often been observed.

Regulations concerning contact during the COVID-19 pandemic required that we conduct the PWI experiment online.

### 2.2. Methods

#### 2.2.1. Participants

A total of 52 native speakers of German (28 male, mean age = 26.81 years, sd = 3.63) were recruited through Clickworker's crowd-sourcing service in Germany (clickworker Europe, Büropark Bredeney, Hatzper Str. 30, 45149 Essen, Germany). Two participants were excluded because they reported further native language(s) other than German; three participants were excluded because we could not hear them speaking in their recordings; and ten further participants were excluded because they did not name the pictures in the intended form (e.g., "The heart is red."). In the end, the data of 37 participants (19 male, 18 female, mean age = 26.76 years, sd = 3.68) were analyzed. Participants received monetary compensation for their participation. All reported normal or corrected-to-normal vision, normal hearing, and normal color vision. The study was approved by the ethics committee of the Deutsche Gesellschaft für Sprachwissenschaft (German Linguistic Society, https://dgfs.de/de/inhalt/ueber/ethikkommission.html (accessed on 10 April 2023)). Written, informed consent, was obtained from all participants prior to testing.

#### 2.2.2. Material

The experimental pictures consisted of 18 colored line-drawings of common objects or animals, scaled to 300 × 300 px. Prior to the main experiment, we had conducted a norming study in which we asked 120 participants to rate on a Likert scale how prototypical/natural

a certain color is for a specific object. Possible judgment values ranged from 1 "not the prototypical/natural color for the object/animal", up to 5 "very prototypical/natural color". Objects and animals were then organized into six color-sets (white, red, black, gold, green, and brown), according to their most highly-rated prototypical color (e.g., *caterpillar*, *salad*, and *cactus* for the set of green-colored things).[4] All color sets comprised three objects each (e.g., *trophy*, *ring*, and *crown* for prototypically gold-colored things).[5] Original pictures were black and white line drawings chosen from a database based on Alario and Ferrand (1999). We then colored the lines in the pictures in the most prototypical color of the object/animal (e.g., gold for a crown), and presented the pictures on a light gray background.

Each target picture (e.g., the picture of a green salad) was paired with a written distractor word: (a) color-related (e.g., *cactus*), or (b) unrelated (e.g., *violin*). All of the distractor words were names of the objects or animals which were depicted in the other experimental pictures. That is, distractor words were part of the response set, however, no color information was included in the distractor words. All related distractor words were re-assigned to different pictures for the unrelated condition. They were semantically and orthographically unrelated to the respective target pictures in both conditions. Of the target's names, ten were monosyllabic, seven were bisyllabic, and one was trisyllabic.

In addition, all target pictures were combined with a context picture, whereby the context picture either introduced an object or an animal of the same color as the target item (narrow focus condition: e.g., *trophy—ring*, both gold-colored objects), or an object or an animal of a different color (broad focus condition: e.g., *crown—tire*, gold colored and black colored, respectively). As in Bergmann and Spalek (2022), we assumed that the presence (absence) of an alternative in the preceding context would license the narrow (broad) focus structure. Context pictures always preceded target pictures during presentation. Thus, in the narrow focus condition, the item pair differed in its object type, but not in its color, and therefore, induced minimal contrast. In the broad focus condition, no such minimal difference was present (see Table 1 for sample items and Figure 1 for examples of pictures used in the experiment).

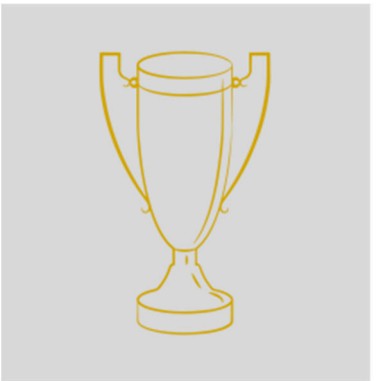 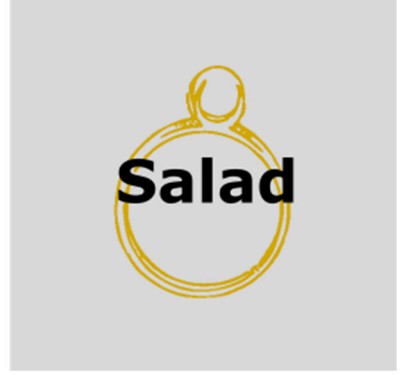

**Figure 1.** Example context picture (**left**) and target picture (**right**).

**Table 1.** Sample items used in the online experiment.

| Condition | Context | Target | Color-Related Distractor | Unrelated Distractor | Colors |
|---|---|---|---|---|---|
| narrow focus | trophy | ring | crown | salad | gold-gold-gold/green |
| broad focus | violin | ring | crown | salad | brown-gold-gold/green |

### 2.2.3. Procedure and Design

The experiment was realized with the online experiment builder PCIbex Farm (Zehr and Schwarz 2018), and was hosted on the server of HU Berlin. Participants received a link to the experiment website, where they read written instructions and performed all tasks

autonomously, without the presence of an experimenter. The utterances were recorded using the recorder element of PCIbex, using the microphone of the participants' choice. In the instructions, we pointed out that a silent environment is needed, and that mobile devices and other applications on the computer should be closed for the duration of the experiment. At the end of each experiment, PCIbex sent the audio recordings, as a zip-file, to the server of HU Berlin.

The experiment was divided into four parts: first, participants were introduced to the experimental items, i.e., the alternative sets. Then, two practice phases followed, before the main experiment started. First, participants were familiarized with all of the alternative sets, as follows: All pictures were presented, grouped by their prototypical color, on the screen, with each group shown for 7 s before the participants could press the "continue" button to see the next group. Captions with target names were written underneath each picture (e.g., 'violin', 'wood', and 'nut', for the pictures belonging to the 'brown group'). The name of the color (e.g., 'brown' for brown-colored pictures), was also written above the group. See Figure 2 for an example of the set-up in the familiarization phase.

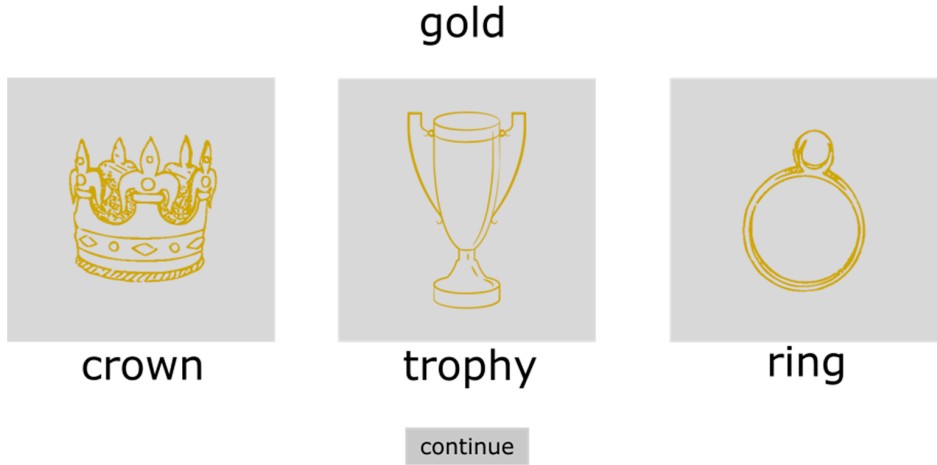

**Figure 2.** Example of the introduction of the "gold" group in the familiarization phase.

After the familiarization phase, two practice phases followed. In the practice phases, pictures were presented in pairs, with the pictures in each pair appearing automatically after one another. The participants were also explicitly informed that the pictures came in pairs. In the first practice phase, half of the 18 pictures were presented, and participants were asked to name each picture in a complete sentence, such as, "The violin is brown", in order to ensure that participants became familiar with the structure of the naming responses (i.e., first the object name, and then the color name). In the second practice phase, the other half of the pictures were presented, while the second picture in a pair appeared together with a superimposed written distractor word that did not occur in the main experiment, i.e., it was not part of the response set (e.g., target: 'The violin is brown.', distractor: *book*). Participants were instructed to ignore the superimposed written words while naming the pictures. The aim of this practice phase was to familiarize the participants with the procedure of the experiment, because no personal examiner could assist during the online experiment. All practice parts were recorded to check post hoc if participants had problems with the procedure.

In the main experiment, we also presented the experimental pictures in pairs: a context picture followed by a target picture. Both pictures had to be named, however, only the latter was shown with a superimposed written distractor word in the center of the picture. Each trial started with a fixation cross presented at the center of the screen for 500 ms. Then the first picture, the context picture, appeared on the screen for 1000 ms, followed by a blank screen. The recording started automatically at picture onset and lasted for 2500 ms. As soon as the recording ended, the target picture appeared automatically,

again for 1000 ms, followed by a blank screen. Another recording started at target picture onset, and ended in 2500 ms. Participants were instructed to: name all of the pictures, in complete sentences, as in, "The violin is brown."; to name them as quickly and accurately as possible; and to ignore the superimposed written words whilst naming. After the second recording ended, participants could press the spacebar to start the next trial. Distractors were presented together with the target picture at three SOAs: −100 ms, 0 ms, and +100 ms, and disappeared together with the picture. Figure 3 illustrates the process of one experimental trial.

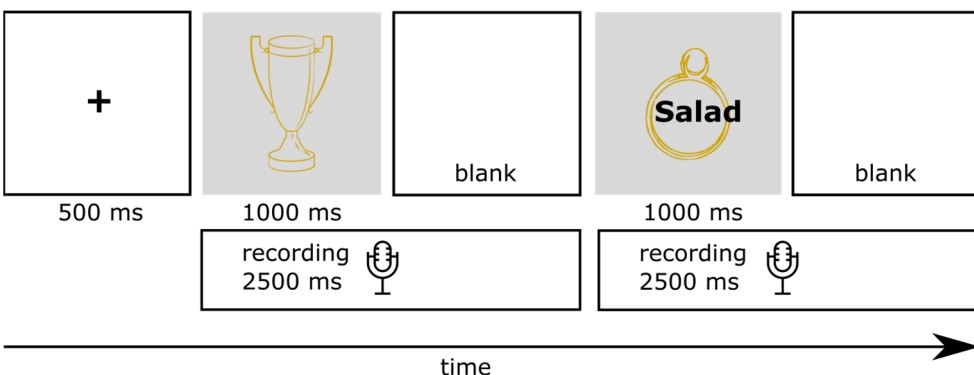

**Figure 3.** Example of an experimental trial. The distractor word appears according to the SOAs on the target picture.

Each target item appeared in all four conditions[6] (narrow-focus-related, narrow-focus-unrelated, broad-focus-related, broad-focus-unrelated), so there were 18 × 4 = 72 context-target-distractor triples for the 18 items. For each SOA, all context-target-distractor triples were presented. Trials with the same SOA were organized into two blocks of the same length, in order to reduce the length of a single block. Each participant received all SOAs. The order in which participants received the SOA-blocks was balanced according to a Latin-square design.

Experimental items within each block were presented in a pseudorandomized order, with the constraint that the same object/ animal was not repeated within four consecutive trials, neither as picture nor as distractor word. Phonological overlap with subsequent items was also avoided.

Altogether, the main experiment comprised 18 (items) × 4 (conditions) × 3 (SOAs) = 216 critical trials, equally distributed among 6 blocks separated by short breaks, and lasted approximately 45 min.

*2.3. Results*

All recorded answers were transcribed and annotated. We measured naming latencies by extracting reaction times from the recordings using Praat. Speech onset was annotated manually in Praat. For all recordings, the annotation of the plosive [d] at sentence onset was placed at the start of the burst visible on the spectrogram. The latency between the start of the recording (picture onset) and the annotation was then retrieved as the reaction time. The naming responses were annotated according to nine categories: "correct" (the participant named the object and color correctly in the intended form, as in "The violin is brown"); "syn" (a synonym of the object was used instead of the intended word); "sem" (a semantically related object was named instead of the target object); "hesitation" (the participant hesitated by saying filler words such as "uh", "erm" before the naming); "self-correction" (the participant corrected themselves while naming); "col" (the participant named the wrong color); "det" (the participant named the object without a determiner); "incorrect" (the participant either named something completely unrelated to the target object, or named the picture in a different form than instructed, such as "the brown violin"); and "noAns" (no response). A total of 7.6% of the trials were discarded due to incorrect or

no responses in the naming task. Trials with response hesitation or self-correction (2.7%), as well as trials with semantic errors (e.g., named semantically related target object words), were also considered to be errors, and were discarded (2.4%).

A further 2.4% of the remaining trials were eliminated because their naming latencies deviated more than 2.5 standard deviations from a participant's mean RT in each condition. Another 1.81% of all correct trials were eliminated after model criticism (see below), because standardized residuals had a distance greater than 2.5 standard deviations from 0.

All remaining reaction times were logarithmically transformed and analyzed with a linear mixed effect model using the R-package lme4 (Bates et al. 2015), and lmerTest (Kuznetsova et al. 2017). The model included participants and target pictures as random effects, while color relatedness, focus, SOA, and their interactions, were included as fixed factors. Fixed factors were sum-coded.[7]

We observed a significant main effect of color relatedness with faster reaction times for the unrelated condition (mean = 847 ms, sd = 195), than the related condition (mean = 863 ms, sd = 204). Reaction times in the narrow focus condition were faster (mean = 848 ms, sd = 194) than in the broad focus condition (mean = 863 ms, sd = 205). The interaction was not significant. However, the data were qualified by significant interactions of color relatedness and focus with SOA. Table 2 presents the model estimates, and Figure 4 illustrates the effect by SOA.

**Table 2.** Fixed-effect estimates (top) and variance estimated (bottom) for LMER of logarithmically transformed reaction times (log(RT) ~ColorRelatedness × Focus × SOA + (1 | Participant) + (1 | Item), n = 6714, REML = −4831).

| Fixed Effects | Coefficient | SE | t | p |
|---|---|---|---|---|
| Intercept | 6.728 | 0.023 | 289.22 | |
| Color Relatedness | 0.008 | 0.002 | 4.04 | <0.001 |
| Focus | −0.008 | 0.002 | −3.91 | <0.001 |
| SOA (−100 vs. 100) | 0.056 | 0.003 | 19.65 | <0.001 |
| SOA (0 vs. 100) | 0.004 | 0.003 | 1.45 | 0.15 |
| Color Relatedness × Focus | −0.003 | 0.002 | −1.54 | 0.12 |
| Color Relatedness × SOA (−100 vs. 100) | −0.006 | 0.003 | −2.14 | 0.03 |
| Color Relatedness × SOA (0 vs. 100) | 0.006 | 0.003 | 1.96 | 0.05 |
| Focus × SOA (−100 vs. 100) | 0.009 | 0.003 | 3.14 | 0.002 |
| Focus × SOA (0 vs. 100) | −0.006 | 0.003 | −2.22 | 0.03 |
| Color Relatedness × Focus × SOA (−100 vs. 100) | 0.002 | 0.003 | 0.53 | 0.59 |
| Color Relatedness × Focus × SOA (0 vs. 100) | −0.003 | 0.003 | −1.21 | 0.23 |

| Random Effects | Variance |
|---|---|
| Participant | 0.0181 |
| Item | 0.0009 |

We analyzed the data separately for each SOA. For SOA −100, there were no significant effects of color relatedness or focus, and no significant interactions (all *p*s > 0.33).

For SOA 0, the effect of color relatedness was significant (B = 0.014, t = 4.31, *p* < 0.001), showing the classic interference effect with longer reaction times for related (mean = 871 ms, sd = 201) compared to unrelated (mean = 843 ms, sd = 178) conditions. The effect of focus was also significant (B = −0.014, t = −4.33, *p* < 0.001), with faster reaction times for elements in narrow focus (mean = 844 ms, sd = 183 vs. mean = 870 ms, sd = 198). The interaction of color relatedness by focus was significant (B = −0.007, t = −2.11, *p* < 0.05). A paired t-test showed that the color interference effect was not significant for the narrow focus condition (t = 1.41, *p* = 0.15), but it was significant for the broad focus condition (t = 3.61, *p* < 0.001).

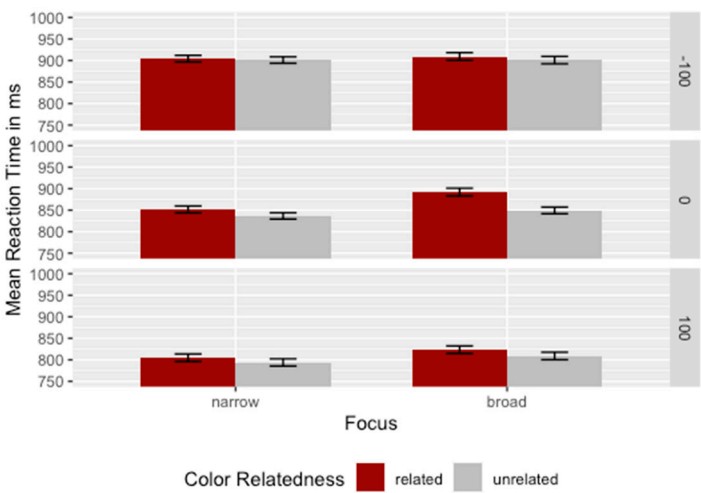

**Figure 4.** Mean reaction times (in ms) for all conditions by SOA.

Finally, for SOA 100, the main effect of color relatedness was significant (B = 0.009, t = 2.49, *p* < 0.05), with longer reaction times for related conditions (mean = 814 ms, sd = 209, vs. mean = 801 ms, sd = 203). The main effect of focus was also significant (B = −0.0010, t = −2.81, *p* < 0.01), with faster reaction times in the narrow focus condition (mean = 799 ms, sd = 205 vs. mean = 816 ms, sd = 207). The interaction was not significant (B = −0.001, t = −0.31, *p* = 0.76).

## 3. Discussion

In the current study, we tested whether the semantic interference effect is affected by focus, that is, through the process of activating focus alternatives. We conducted an online PWI experiment and presented related sets of elements. Their relatedness, however, was not based on co-hyponymy, but rather on sharing the most natural/prototypical color (e.g., *salad*, *caterpillar*, and *cactus* all share the prototypical color *green*). Pictures were named with small sentences, as in, "The salad is green". Focus was licensed by context pictures preceding the target picture (e.g., *salad*). Context pictures showed an object of the same alternative set (narrow focus), or an object of a different alternative set (broad focus). We varied the SOA to investigate the time course of focus alternatives activation.

Results showed faster response latencies in the unrelated condition compared to the (color) related condition at SOAs 0 and 100, reflecting the classic semantic interference effect (see e.g., Damian and Bowers 2003). This successful replication is remarkable for two reasons. First, the present study shows that language production studies can be carried out online, even though the experimenter has less control over the technical apparatus. Second, it is one of the few studies that observe interference in PWI for target and distractor words that are not co-hyponyms.

Abdel Rahman and Melinger (2011), for instance, showed that otherwise unrelated elements can be co-activated when a meaningful link (e.g., a theme or title) licenses their semantic connection. Providing such a contextually meaningful connection, they found interference in blocked naming experiments with elements that did not share a common hypernym. Alternative sets in our online experiment were connected (and familiarized) in a similar way, that is, by means of natural color-categories, which licensed co-activation during picture naming. The co-activation of elements sharing the same prototypical color, could lead to lexical competition during the picture naming process.

While PWI is usually used to investigate production of single words (but see Schnur et al. 2006), the results further demonstrate that the classic interference effect also occurs at the sentence level at which focus also applies.

An interaction of focus and color relatedness was only significant at an SOA of 0 ms. In detail, color relatedness affects only broad focus, but not narrow focus. Instead

of the predicted increase in interference, we observed the opposite: that the interference effect was reduced for narrow focus compared to broad focus. In general, the effect of co-activation of focus alternatives on language processing can take many different forms, depending on the languages investigated, the manner of focus marking, the type of baseline, and the task employed. In language comprehension, Gotzner et al. (2016) observed for German that the recognition of mentioned alternatives is harder when focus is marked with particles. Calhoun et al. (2023) report similar findings for Samoan, where focus is marked by cleft-like structures, as do Káldi et al. (2021) for the preverbal focus construction in Hungarian. By contrast, Yan and Calhoun (2020) found that prosodic focus marking in Mandarin Chinese and in English, facilitated the decision that an alternative had not been mentioned. Thus, at the moment, while we are still beginning to understand the complex interactions of different factors in the processing of focus, both facilitation or interference can be compatible with an account where alternatives are co-activated, while a focused element is being processed.

In order to provide a post-hoc explanation for the direction of the effect in our data, we turn to some observations from language comprehension from our lab: Spalek and Oganian (2019), and Joerdens et al. (2020), both investigated probe recognition times of single words after a spoken sentence, with narrow focus either on the object or the subject. Participants had to decide whether the word had occurred in the previous sentence or not. If the focus was on the sentence subject, a probe word that was semantically related to the sentence object was very hard to reject, compared to an unrelated word. By contrast, if the focus was on the sentence object, participants found it very easy to say that an alternative to the object had not occurred in the sentence. This pattern fits well with the present finding, where a difference between unrelated and related distractors was present in the broad focus case but not in the narrow focus case. It is as if focus helps to very clearly distinguish the target word from all possible distractors, thereby decreasing inhibition (see also Sturt et al. 2004, for comparable findings).

According to Levelt (1989), the assignment of topic and focus to a to-be-produced utterance, precedes lexical access in language production. Therefore, a concept that is the designated focus might send an extra boost of activation to its lemma. This would imply that the target word is more active relative to co-activated competitors, and therefore easier to select. It remains an open question why the time window where this happens seems to be narrower than the time window for lexical competition. However, in the early days of research on lexical selection, researchers also narrowed down the relevant time windows very much through trial-and-error, so our finding is a promising starting point for future investigations.

One caveat concerns the relatively low number of participants. We had aimed for 52, but due to the lack of control in online experiments, only 37 could be analyzed. Given that the present study is the first to investigate lexical access during the production of focused elements, there were no relevant studies that we could use in order to determine a valid sample size. Thus, it could well be the case that the critical interaction of focus and color relatedness might be significant for SOA 100 ms, if a larger sample is tested. Again, we think that future investigations need to follow from the present study, including replications that confirm the present findings.

It is also possible that not all participants were aware of the underlying focus structure. Controlling for this is less easy than it seems: an obvious solution would be to analyze only the data for those speakers who produce the focused element with contrastive intonation. However, there is no simple way to determine whether an individual has used contrastive focus. The mapping of a particular pitch accent to a focus type is not clear-cut, and varies individually: Grice et al. (2017) collected data from 5 participants producing the same words in broad, narrow, and contrastive focus conditions. Not all speakers showed the expected categorical shift from H+!H* to H* to L+H* for broad, narrow, and contrastive focus, respectively. In addition, Cangemi et al. (2015) have shown that listeners, too, are highly variable in detecting prosodic contrasts for different types of focus (broad, narrow,

contrastive). Most importantly, while a listener X can reliably detect differences between speakers A and B, but not C and D, a listener Y can detect the differences between speakers C and D, but not A and B.

The findings discussed above suggest that determining the focus structure of a single trial in a production experiment is a challenge: neither automatized scripts nor a human listener will pick up all correctly produced prosodic contrasts. Therefore, we decided to focus on reaction times only, and not to undertake any acoustic analyses. Nonetheless, the interaction of the PWI effects with the actual focus structure produced by the participants, is an important question that needs to be tackled by future research—preferably in a well-controlled recording environment, and not in an online study where every participant used their own microphone.

To conclude, with the current online experiment, we can show that elements connected through a common natural prototypical color, cause interference. Further, focus affects color interference such that the interference decreases at an SOA of 0 ms, but not at the later SOA of 100 ms. The precise dynamics of the interplay between color interference, which was present for SOA 0 and 100, and the modulating effect of focus, which was only present at SOA 0, will have to be further investigated in future experiments.

**Author Contributions:** Conceptualization, B.B. and K.S.; methodology, B.B. and Y.L.; formal analysis, B.B. and K.S.; data curation, B.B. and Y.L.; writing—original draft preparation, B.B.; writing—review and editing, K.S.; visualization, K.S. and Y.L.; supervision, K.S.; project administration, K.S.; funding acquisition, K.S. All authors have read and agreed to the published version of the manuscript.

**Funding:** The research was funded by the European Union's Horizon 2020 research and innovation program under grant agreement No GAP-677742 awarded to K.S. The article processing charge was funded by the Deutsche Forschungsgemeinschaft (DFG, German Research Foundation)—491192747 and the Open Access Publication Fund of Humboldt-Universität zu Berlin.

**Institutional Review Board Statement:** The study was conducted in accordance with the Declaration of Helsinki, and approved by the Ethics Committee of the German Society for Linguistics (Deutsche Gesellschaft für Sprachwissenschaft) (Antrag 13, date of approval: 21 August 2015).

**Informed Consent Statement:** Informed consent was obtained from all subjects involved in the study.

**Data Availability Statement:** Data are not available publicly since no consent was obtained for publishing participants recordings. A data table with transcriptions of participants' errors can be obtained from the authors upon request.

**Conflicts of Interest:** The authors declare no conflict of interest.

## Notes

[1] Capital letters indicate contrastive accent.

[2] The original stimuli were presented with a double focus condition ("My SON likes SPINACH") because focus on just the final noun could be mistaken as main sentence stress, rather than contrastive focus.

[3] The condition labels do not depend on how speakers actually produce the utterance. We reasoned that the context (and therefore the focus condition) was present in the linguistic material, i.e., the sequence of sentences, even if participants chose to not mark this prosodically.

[4] Whereas some items have a single prototypical color, others allow more variability (e.g., *books*). We assigned items to categories according to the color that was chosen as the prototypical one most often in the norming study. In addition, during training (see Section 2.2.3. Procedure and design), participants learned which color an item had in our study.

[5] We used alternative sets based on shared features, because the use of co-hyponyms had led to a confound that cannot easily be avoided (for details, see Bergmann and Spalek 2022). In addition, having used color-related items in cross-modally primed LDT, we wanted to replicate the finding that these items can be alternatives to one another.

[6] Repeating items in these types of experiments is quite common and was also undertaken by: Costa et al. (2005); Damian and Bowers (2003); Janssen et al. (2008); Mahon et al. (2007); Schnur et al. (2006); Schriefers et al. (1990).

[7] Sum coding a contrast in a linear mixed effects model, means that the intercept represents the grand mean, which resembles the coding of traditional ANOVA analyses.

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
