# Peer review of "The Influence of Focus on the Activation of Alternatives in Speech Production—An Online Picture-Word-Interference Experiment"

_languages, doi:10.3390/languages8020110_

Round 1
Reviewer 1 Report
see attachment

Reviewer 2 Report
Review of Article “The Influence of Focus on the Activation of Alternatives in Speech Production – An Online Picture-Word-Interference Experiment”
Summary
This paper reports on a production study in German that investigates the “semantic interference effect” at sentence level in order to explore (a) whether interference effects can also be evoked when the relatedness is not based on semantic categories, but on sharing the most natural prototypical colour (colour-relatedness), (b) whether interference effects are affected by the focus domain (broad vs. narrow) and (c) the time course of focus alternative activation during speech production. The study is a logical follow-up of a previous lexical decision task (in press) from the same laboratory and uses the same (or similar?) material for a picture-word-inference experiment which was conducted online. Participants had to name a context picture and a following target picture by short descriptive sentences, e.g. “The heart is red”. Focus on the target sentence/noun was modulated by the two pictures differing in object type, i.e. object or animal, and colour (broad focus) or in object type only (narrow focus). The target picture was presented together with a written distractor word (an object/animal either related or unrelated in colour) at three stimulus onset asynchronies (SOAs: -100 ms, 0 ms, 100 ms). Interference effects were interpreted on the basis of naming latencies (reaction times) measured in relation to the picture onset. The results overall reveal faster reaction times (i.e. less interference) for i) the unrelated than the colour-related condition (classic semantic interference effect) and ii) the narrow than the broad focus condition. These effects have also been found for SOA 0 and SOA 100 plus an interaction of relatedness and focus for SOA 0. These findings have important implications for our understanding of the activation of alternatives in speech production, since they seem to suggest that relatedness can also be licensed by a contextually meaningful connection (here the colour category), at least in a broad focus domain.
General Concept Comments
On the whole, I liked the paper. It is well-motivated and addresses interesting, and as far as I can see, relevant and original questions in the study of focus alternative activation and semantic interference. The experiment itself is well designed, the arguments are reasonable, and the theoretical background and interpretation of the results appears, overall, sound. However, since the methodology of the experiment is quite complex and the presentation is quite dense, I think clarity, detail/transparency, and consistency could be improved in some places to facilitate a better understanding (also in terms of reproducibility). In light of this, I have (apart from minor issues) five main issues (outlined below) that I feel need to be addressed prior to publication. Overall, this is a case of “minor revisions required” and I am pleased to recommend the paper for publication.
1) Research objectives:
- If I have understood it correctly, you actually address three research questions/objectives in your study, as I already formulated in the summary above: (a) the effect of colour-relatedness, (b) the effect of focus domain, (c) the time course of focus alternative activation. In view of the title of the paper, I would expect that (b) is the main research objective. While reading the paper, however, I got the impression that (a) is the main objective since you repeatedly mention and highlight the investigation of colour-relatedness (“the most natural prototypical colour” as shared feature). The difference in focus domains of question (b) rather seems to serve as a control parameter, at least you are introducing it as such (see p. 4, l. 177), and the role of (c) seems to be somewhat arbitrary since SOAs are mentioned not before section 2 (p. 4, l. 146). Please make clear right from the beginning what exactly your research questions/objectives are and if possible also try to rank their importance/role for this paper.
- Moreover, different research questions and aims are introduced in different sections, i.e. at the end of section 1.4. (p. 4, l. 149) and in section 2.1. (starting with SOAs) which contributes to the lack of clarity just mentioned above. I would have expected a brief overview of all research questions (ranked according to their importance) in section 1.4. and a detailed explanation of all questions (in the same order) in section 2.1. Hence, you might also want to rethink the structure and place for introducing the research objective/questions.
2) Hypotheses:
- In line with the formulation of the research questions, I find the hypotheses (p. 4, l. 173-178) rather vaguely formulated. I was wondering whether your study is on the whole meant to be exploratory, i.e. without clear hypotheses (as formulated on p. 4, l. 147-148: “Hence, the questions arise as to when focus alternatives become available during speech production processes and whether they cause interference or facilitation”) or has a confirmatory (hypothesis-testing) part related to research questions (a) and (b) and an exploratory part related only to research question (c). Please make clear (if applicable) which parts of your study are confirmatory and which are exploratory and formulate clear hypotheses for each (confirmatory) research question.
- In section 2.1. you assumed to find a stronger interference effect for related words in the narrow focus than in the control (broad focus?) condition (cf. p.4, l. 175-177). However, you found the opposite result and this is not taken up and discussed later in the Discussion section (if I see it correctly). Please make sure that you discuss your results with reference to the formulated hypotheses.
3) Procedure:
- On p. 6, l. 232-236 it says that participants were tested individually and in the Discussion it says that “… the experimenter has less control over the technical apparatus” (p. 9, l. 347). Since the study was conducted online, this makes me wonder whether and if so how there was an experimenter (e.g. via video conference) or whether the participants did the study completely autonomous. Please provide more details about i) how you conducted the experiment ii) how the utterances were recorded (via open broadcaster software, cell phone, what types of microphone?) and iii) about an ethics statement.
4) Colour-relatedness:
I am afraid, I am confused about your classification of the colour-relatedness feature: At the end of the Introduction and also at the beginning of the Discussion you argue that this feature is NOT based on SEMANTIC relatedness (see p. 4, l. 152-154 and p. 8, l. 334-336). However, in contrast to this, in your statistical analysis you refer to this feature/factor as “semantic relatedness” (see p. 5, l. 302). Moreover, during the Discussion you bring yet another term into play, i.e. “taxonomic category”. Referring to this, you argue that the colour feature is “semantically connected” (see p. 9, l. 350-352 and p. 9, l. 357-359). At the end of the Discussion, you then conclude that your colour feature is NOT based on a TAXONOMIC relatedness (see p. 9, l. 389-391).
My question now is: Do you suggest that the colour feature is based on semantic/taxonomic relatedness or not? Please clarify this (and also the relation to the SEMANTIC interference effect) already in the Introduction and be consistent in your argumentation. Please also explain what the difference between semantic and taxonomic relatedness is (if there is any). Would it make sense to use the terms “lexical” vs. “conceptual” semantics? For the statistical analyses and the results, I would suggest to use the neutral term “colour-relatedness” (which you already introduced in the Material section; see p. 5, l. 211-212) instead of semantic relatedness.
5) Definition of focus and its relation to prosody:
- In general, I am missing a clear definition of focus and alternatives and an introduction of the different focus domains and types, as the terms “narrow”, “broad” and “contrastive” are relevant for this paper (even if this is a special issue on focus, I would at least expect a short introduction). Moreover, at the beginning of the Introduction, focus is equated with accentuation (see p. 1, l. 21-22). This representation of the relation between focus and prosody is far too simple. Since you are also presenting some studies that are dealing with the relation between focus and contrastive accentuation in section 1.3., it would be helpful to add some more information (and literature) on prosodic focus marking.
- In light of this, I was wondering If you also investigated the prosodic marking (i.e. the accentuation of the noun and the colour) of your context and target sentences? You excluded semantic errors, but what about prosodic errors? Have you considered that a marked/“correct” or unmarked/“incorrect” prosodic realisation of the context and target sentences might also have an effect on the naming latencies. I am aware that a prosodic analysis would go beyond the scope of this paper, but it would be useful to discuss the issues that might arise from the prosodic marking in the Discussion section.
Specific Comments
- p1. ff., 1. Introduction: Please always indicate which language was examined in the presented studies (see in particular section 1.3.).
- p. 1, l. 10: Please do not only use the abbreviation “SOA 0” here, but say what it means.
- p. 1, l. 19: What do you mean by “different pronunciations”? I assume you mean “prosody”/“prosodic realisation” here. Please be a bit more precise.
- p. 1, l. 36: The dot at the end of the sentence is missing.
- p. 2, l. 42-43: The literature mentioned in the parentheses is ordered alphabetically, but later on the order is chronologically. Please, check for consistency.
- p. 3, l. 131: Replace “wide focus condition” with “broad focus condition”.
- p. 4, l. 164: What are the reasons for the choice of the three SOA values? Is there any evidence for these thresholds to be significant in the activation of focus alternatives? Please elaborate on that and introduce the matter of different SOAs already in the Introduction section.
- p. 4, l. 177: Replace “control condition” by “broad focus condition”. This is the only time you mention a control condition, which is not necessary if you only investigate narrow vs. broad focus.
- p. 4, l. 183 and l. 189: Please add the Standard Deviation for the age distribution.
- p. 5, l. 192 ff.: The information about the color norming study comes out of the blue. Please also introduce that earlier and give a reason.
- p. 5, 2.2.2. Material: Please provide example pictures to make it more descriptive.
- p. 5, Table 1: I think in the narrow focus condition in the colours column “black” needs to be replaced by “green”.
- p. 5, footnote 3: I am afraid, I don’t understand this. Could you please rephrase this or elaborate a bit on this?
- p. 6, l. 237-238: How did you introduce the experimental items to the participants (pictures, words)?
- p. 6, l. 283: Please specify how exactly this number of critical trials is calculated.
- p. 7, l. 286-287: Could you give some more information about the annotation of the naming responses and your measurement of the reaction times please. Did you relate the start of the picture to the onset of the speech wave and if so, did you take into account the (non-visible) closure phase of the (sentence initially de-voiced) “d” in the definite article of the context and target sentences?
- p. 7, l. 286-301: This part could better be an Analysis section on its own.
- p. 7, l. 301: Pleas explain what “sum-coded” means.
- p. 7, l. 313: This sentence looks like a figure caption, but it probably just refers to Figure 1. Maybe it makes sense to shift this sentence to the previous or following paragraph, so that it does not get lost.
- p. 8, l. 319 ff.: When you report a significant effect, please specify the effect with regard to differences in reaction times. Please also make clear from the very beginning of the presentation of the results how you interpret the “classic semantic interference effect”, i.e. longer reaction times indicate more interference (or shorter reaction times indicate less interference) and try to be consistent in the presentation of the effect.
- p. 8, Figure 1: The timing differences in the Figure are hard to see. I would suggest to zoom in a bit by reducing the width of the scale of the y-axis.
- p. 8 ff., 4. Discussion: In my view this section is a mixture of Discussion, Introduction and Conclusions. The first and last paragraph give an overview of the study and a summary of the results and therefore should be part of an additional Conclusions section. The part in-between is mostly Discussion, however the topic of co-activation of elements of the same taxonomic category (see p. 9, l. 350-364) should have been presented in the Introduction already as a basis for possible hypotheses for your study.

Round 2
